# Roads to the Sky: Indic Ritual Elements in the Vietnam-China Borderlands and Their Maritime Transmission

David Holm

Department of Ethnology, National Chengchi University, Taipei 11605, Taiwan; dholm@nccu.edu.tw

**Abstract:** One of the basic features of shamanic rituals cross-culturally in East and Southeast Asia is that the ritual itself is structured as a journey up to the sky, climbing the world mountain or the world tree, or else a journey down to the bottom of the sea and back again. The shamanic retinue is understood to make this journey in person, rather than, as in Daoist ritual, sending divine emissaries up to the highest heavens. The journey is conducted through narrative song and dance, accompanied by strings of bells and lutes. The point of departure is the physical village or village household where the ritual is being conducted, and the journey progresses through a series of well-marked way stations via the temple of the earth god to the higher hills and finally to the villages and markets in the sky, before crossing the heavenly seas and ascending the highest mountain. On the way, demons and other impediments are encountered. The route and way stations vary depending on the purpose of the ritual and the intended divine recipient of offerings and submissions. The present article will explore the route up to the sky and the way stations in more detail, taking a single ritual type as performed by the Pụt and Then ritual practitioners as an example. The Pụt and Then are literate ritual specialists found among the Tày and Nùng peoples in northern Vietnam and southern China, near an area which is known to have been a centre of Brahmanical and Buddhist learning from very early times.

**Keywords:** shamanism; shamanic journey; Tày; ritual performance; Vietnam; Guangxi; maritime transmission

## 1. Introduction: Reflections on Shamanism

Rituals of the Tai-speaking ritual specialists called Pụt and Then in northern Vietnam and the southwestern corner of Guangxi in southern China are shamanic in nature, with a basic structure that is South Asian in origin and came to the Red River Valley area via the maritime routes from the Indian subcontinent, probably in the late centuries BCE or very early CE (see Holm 2018, 2019). I define shamanic rituals as rituals which are structured as journeys up to the sky or down to the bottom of the sea, with shamanic practitioners and their retinue making the journey themselves, with songs and dances marking or rather instantiating the stages in the journey. Typically, the retinue includes soldiers and horses, but also porters to carry the offerings and rowers to row the boats across the celestial seas. The present article will review those earlier findings, provide additional information on key points, and then explore the stages in the road to the sky in more detail. It can be shown that many stages in this journey are indigenised and covered over with a layer of place names and names of deities which are either Chinese or local Tai, but that others, including the basic cosmological framework, remain unmistakably Indian.

I would argue that these rituals are fundamentally different from those conducted in the same region by spirit mediums, which involve calling deities or departed ancestors down into the body of the practitioner, followed by the practitioner being taken over psychically by the deities or ancestors and speaking and acting as their mouthpiece. Such rituals are common among the ethnic Vietnamese communities in northern Vietnam, where they are known as *lên đồng* 'ascending a child-medium'.[1] Similar practices are also widespread in the

southern part of China (see e.g., Davis 2001, pp. 87–199). The direction of movement in these rituals is exactly the opposite of shamanic journeys, with gods and spirits descending into the body of the medium. Also fundamentally different is the degree of control the practitioner has over the interactions with spirit entities: with spirit mediums the spirits are in control, while in shamanic ritual the shaman has a considerable degree of control over interactions with spirit entities.[2]

As is well known, I.M. Lewis in his *Ecstatic Religion* argued that the Tungusic evidence from northeast Asia made nonsense of the idea that there was a distinction between spirit mediumship and shamanism, 'belonging necessarily to two different cosmological systems and to separate stages of historical development' (Lewis 1971, p. 56). It is worth pointing out here that more recently Edward Davis has roundly refuted that argument (Davis 2001, pp. 2–3).

Michel Strickmann in his posthumous book *Chinese Magical Medicine* (Strickmann 2002) noted some of the confusions caused by referring to spirit medium and Tantric Buddhist-derived ritual elements as 'shamanism'. He points out that Carmen Blacker's study of 'fox families' in Japan (Blacker 1975) was actually a study of Tantric-derived ritual practices (p. 271), and comments that her use of 'shamanistic' in the subtitle 'seems both to weaken the potential usefulness of the term "shamanism" and to blunt or even efface the historical significance of the Buddhist contribution'. Further on (p. 279), referencing Eliade's work and 'archaic techniques of ecstasy', he quite rightly points out that 'the historian is obliged to note the presence of elements derived from the dominant ritual patterns of organized "higher" religions', and goes on to say, 'For some twelve centuries, the dominant ritual patterns in much of East Asia have been those of Tantric Buddhism and Taoism. The term "shamanism" is convenient, to be sure, but perhaps too convenient. It is certainly too slippery and imprecise to be used without qualification in complex societies like those of China and Japan'.

When applied to Tantric-based practices of spirit possession, of course, Strickmann's argument has considerable point. But the comment about the term 'shamanism' being 'too slippery and imprecise' is based on a narrow focus on Tantric Buddhism and Taoism, either ignoring or remaining ignorant about the ritual practices of pre-Tantric Buddhist ritual, in the form that it was first introduced to East and Southeast Asia. What we will be investigating in this article is a form of ritual practice that arguably predates the development of Tantric or esoteric Buddhism in India and its subsequent transmission to East Asia by some centuries.[3]

There is one point raised by Strickmann that is interesting and potentially useful, however. In both books he points out that the first authors to describe North Asian shamanism proposed that the term "shaman" derived from the Sanskrit term *śramaṇa*, meaning 'Buddhist priest', and comments that this hypothesis was one that had never been categorically refuted.[4] Actually, the Chinese term 沙門 *shamen* appears in the very earliest Buddhist scripture in Chinese, the (*Fo shuo sishi'er zhang jing*) 佛説四十二章經, which is reckoned to date from the 1st century CE.[5] Even more intriguingly, Zheng-Zhang Shangfang's Old Chinese reconstruction of these two characters 沙 'sand' and 門 'gate' is OC \*sraal and OC \*mɯɯn (Zhengzhang 2003, pp. 454, 418), which gives us a strikingly close match with the Sanskrit *śramaṇa*. This may be fortuitous, of course.

There are other, broader issues which would merit further clarification. Journeys to the sky are based on the Vedic ritual *vājapeya*, the rite of ascension. Eliade discusses this ritual in several places in his text,[6] and this can be seen as one of the sources of rituals structured as journeys to the sky (Holm 2018, p. 30). Indeed, a good many of the examples in Eliade's classic work on shamanism are shamanic rituals of this type.[7] However, there are two points in his overall framework which need to be challenged. The first is his identification of trance as the defining feature of shamanism. There are other scholars, some of them quite eminent, who have challenged this identification.[8] As I have noted in a previous paper, trance is a phenomenon found in rituals throughout the world: in North America, in South America, in Africa, and so on, places with no plausible connection historically with

religious practices in East or Southeast Asia (Holm 2018, p. 4). Its appearance in what we might call primordial religious practices would seem to be almost universal.

Admittedly, the use of trance is not entirely absent from the rituals of the Then and Pụt in northern Vietnam: there is a ritual segment involving collective trance that has been documented for their ordination ritual, but it plays a minor part (Nguyễn Thị Yên 2010, pp. 94–95; cited in Holm 2019, p. 8). The primary mechanism for travel to the sky, however, is Brahmanical-style recitation of the words of the ritual text, which have the power to make real the events and places referred to. The psychic concomitants of this spiritual journey in the minds of practitioners have yet to be thoroughly investigated, but as we shall see, at least it is clear from their recitations that both the Then and Pụt are accompanied on this journey by the spirits of their ancestral masters. Elsewhere along the China-Vietnam border, it is not unknown for some Tai-speaking ritual specialists to combine shamanic journeying and spirit medium-based practices within the same ritual, though again the journey and the direct communication from ancestral and other spirits are found in separate ritual segments, clearly demarcated (Kao Ya-Ning 2009, pp. 207–22).

In Eliade's classic study, the choice to identify trance as the defining feature of shamanic ritual allows him to include also the practices of spirit mediums and ritual specialists worldwide. This is unhelpful and is one of the things that have led to the confusions noted above. Of course, Eliade was writing at a time when there was less information about the timing of the Indian incursions into the Red River Valley and elsewhere in Southeast Asia, about the presence of a strong Vedic and Brahmanical component in these early ritual practices along with Buddhism, and about the relatively late development and introduction of ritual practices based on Tantric Buddhism some centuries later. In a number of places in his classic work, he notes definite signs of influence from more complex societies on the rituals of Southeast Asia but declines to provide an historical narrative to account for the facts on the ground.[9] On this point he was being over-cautious: he draws the bow without releasing the arrow. All the evidence he marshals indicates that rituals structured as journeys to the sky or down into the underworld came from the Indian subcontinent and were part and parcel of the assemblage of Brahmanical ritual templates introduced into this part of the world in the early centuries of the common era. Of course, providing such an 'historico-cultural' explanation would have put paid to the idea that shamanism was a human universal with trance as its defining feature.

The discussion here needs to be set against a background in which the term "shamanism" has been given a very wide range of meanings in international scholarship. Earlier generations of scholars like Piet van der Loon used the term 'shamanic substratum' to refer to the non- or pre-Confucian bedrock of Chinese culture and society (van der Loon 1977). In the study of Chinese religion, "shamanism" has also sometimes been used to refer to any local practices that lie outside the orthodox traditions of Buddhism, Daoism, and Confucianism, or as a vague term corresponding to Chinese *wu* 巫 (Lagerwey and Marsonne 2015, p. 15, fn. 13). Elsewhere, shamanism has been used as a term to refer to the indigenous religious practices among a wide range of first peoples in the Americas as well as in Africa and other places. One might note that used in this sense, the term still has some connection with Eliade's definition, and at least is not pejorative in the way that terms like 'witch-doctor' or 'ghost-woman' are. Of course, used like this, it may also be thought to be a way to incite readers' interest in primordial wisdom. My own preference would be to use terminology which distinguishes clearly between all these different forms of ritual practice.

## 2. Tày and Nùng

The Tày and Nùng are Tai-speaking peoples living on both sides of the China–Vietnam border. Both groups practice wet-rice agriculture in flat-bottomed mountain valleys. Their habitat is distinct from that of hill-dwelling peoples such as the Miao (Hmong) and Yao. In Vietnam the Tày are the minority group with the largest population nationally, numbering some 1,626,392 people in 2009.[10] The Tày are now concentrated mainly in the northern

provinces of Cao Bằng, Lạng Sơn, Bắc Kạn, Thái Nguyên, Quảng Ninh, and Bắc Giang, where they have lived for several millennia.[11] In Vietnam the Tày are officially classified separately from the Nùng, who are also Tai-speaking but are descendants of relatively recent migrants from China. In Guangxi, however, the Nùng (*bouxnungz*) are classified together with the Zhuang, as are the Tày, although the Tày there refer to themselves as Bu-Dai 布傣, a transliteration of the native term.

In Vietnam there are a good many Nùng groups, who are said to be distinct from one another in language and customs. They are usually referred to by their places of origin in China, which are likewise different. Thus Diguet, in his treatise on mountain peoples of the borderlands, gives the following list of 12 different groups for the province of Cao Bằng (Diguet 1908, p. 68):

> 1 Nung Hin, speaking an idiom resembling that of the Thô group. Their origin is the châu of Lung Hin, northwest of Tai Ping Fou.
>
> 2 Nung Han, whose dialect and customs are quite distinct from those of the other groups. They came from the châu of Lung Han.
>
> 3 Nung Loi, who resemble the Nung Hin and have arrived recently from Tonkin.
>
> 4 Nung Châu, whose language closely resembles that of Thô and who originate from Long Châu.
>
> 5 Nung Kenh Lai, whose language and customs are similar to those of the afore-mentioned groups.

No information about place of origin is given about the other groups (6–12), which include the Nung Phan Sênh, the Piang, the Giang, the Nung An, the Min, the Ngan, and the Gioi.

Most of the places of origin mentioned above are located in the southwestern part of Guangxi, an area in which predominantly Southern Zhuang (Central Tai) languages are spoken.[12]

### 3. Pụt and Then

Among the Tày and Nùng in Vietnam and the southwestern borderland of China there is a variety of different kinds of religious practitioners performing different functions for communities. The Pụt and Then concentrate on rituals for the benefit of the living, and recite texts in the Tày or Nùng language along with ritual songs of offering.[13] Both the Pụt and the Then perform rituals which involve shamanic journeys into the sky. It is significant that the names Pụt and Then are both usually written with the same character in their texts: 伏, which is a vernacular variant for 佛 *fo* 'buddha'.[14] With the designation "Then", the character 伏 has been re-interpreted, so that 天 *tian* 'heaven' on the right-hand side is understood as indicating the pronunciation. But it is more complicated than this, for 'Then' is also the general designation in the Tai-Thai languages in Southeast Asia for sky gods. And sky gods, in turn, represent a particularly well-developed and powerful sector of the Tai pantheon, being closely connected not only with harvests and general prosperity but also with political power and chieftaincy in the Tai domains (see Archaimbault 1991).

There are differences in performance style between the Pụt and Then, which Bế Viết Đằng discusses in some detail (Bế Viết Đằng and Lục Văn Pảo 1992, p. 167). Overall, Then lyrics are often based partly on older Pụt material, but the Then represent a form of performance which is artistically more developed. There is, however, quite a lot of overlap in the content of their ritual manuscripts, and the Then tend to be referred to in quite a number of localities as 'Pụt'.[15] They share much of the same repertoire.

Both Then and Pụt have both male and female practitioners, who often perform together during rituals. This is significant in itself, and quite unlike the situation in most of Guangxi, where male and female specialists may collaborate and take part in the same rituals, but have quite different ritual traditions, often even in different languages.[16]

## 4. Literature Review

Rituals of the Tày and Nùng Then and Pụt are copiously documented in Vietnamese scholarship, and there has been some translation of this material into French and, more rarely, English. There are bilingual editions of the texts of almost all the major rituals conducted by the Then. These editions include a transcription of the manuscript reading in romanised Tày, a Vietnamese translation of the Tày text, ethnographic and textual notes, and often a photo-reproduction of a Tày manuscript or a transcribed version of one in vernacular character script.[17] Such publications are available for the following Then rituals:

1. The Ordination Ritual (Lễ Cấp Sắc) (e.g., Nguyễn Thị Yên 2007)
2. Ritual of Praying for a Harvest (Lễ Cầu Muà, Lễ Hội Lồng Thồng)
3. Praying for Peace (Lễ Kỳ Yên) (e.g., *Tổng tập truyện thơ nôm các dân tộc thiểu số Việt Nam* 2013)
4. Praying for Progeny (Lễ Cầu Tự)
5. Removing Astral Impediments (Giải Hạn) (e.g., *Tổng tập truyện thơ nôm các dân tộc thiểu số Việt Nam* 2012)
6. Celebrating Longevity (Lễ Cầu Trường Thọ, Then Chúc Thọ)
7. Visiting the Goddess in the Moon (Lễ Hội Hằng Nga)
8. Playing with the Swallows (Then Hỉn Ên)

In addition, an entire ritual by the Tày Pụt and accompanying photo-reproduction of manuscript pages has been published in a trilingual Tày–Vietnamese–French edition, with relatively copious notes and a lengthy introduction by Tày scholars. The manuscript itself, written in the Tày vernacular character script, was in the possession of a man called Hoàng Quang Ngọc in the commune of Đông Phục in the southern part of Ba Bể district, which was then part of Cao Bằng province. This text included 2832 lines of the 3030 lines included in the published volume; the material added came from a manuscript in the possession of Nông Văn Phia of Cao Thượng commune, also in Ba Bể district, and four passages of missing material which another Pụt in the region, La Đinh Sôi of Phương Viên commune in nearby Chợ Đồn district, was able to recite by heart.[18]

Some of the above ritual categories are communal rituals, performed on behalf of a community with the participation of as many as a dozen practitioners, and others are household rituals performed for families or for individuals, with often just one priest conducting the ceremony. Household rituals are conducted in the main hall of the family house, in front of the spirit shelf where the ancestral spirits are located, while communal rituals are conducted in an open space outside in the village.[19] Most of these ritual texts are of very considerable length: the 'Celebrating Longevity' text is 5775 lines long; the 'Praying for Peace' ritual takes an entire night to perform. They are divided into a number of songs or sections, often labelled as such with subtitles in the manuscripts.

My analysis of this vast corpus of Tày shamanic songs and texts is at a very early stage. What I have done thus far is to prepare interlinear editions with word glosses and Tày-English glossaries for two versions of the 'Crossing the Seas' segment; interlinear transcriptions for 'Praying for a Harvest' and 'Removing Astral Impediments' manuscripts; surveyed the ways in which a list of 60 common words were written for some 10 Tày and Nùng manuscripts in different locations; and written two articles on reflections of voiced initials in the vernacular Tày script, using the 'Crossing the Seas' text as an example.[20]

In Vietnam, scholarly investigation of the Then got off to a very early start, during the final stages of the war with the US. This was well before Đổi Mới the Vietnamese reform period beginning in 1986. The survey and investigation process is described in admirable detail in a conference volume 'Some Questions on the Then of Northern Vietnam' (Mấy vấn đề về Then Việt Bắc), published in 1978.[21] Surveys and documentation on the Then were motivated in the first instance by the important role Then songs and Then practitioners had played in the war effort and in popular education drives among the general populace in what was then the North Vietnam Autonomous Region, a region that comprised five provinces in the northern area that had majority Tày and Nùng populations. Traditional

Then songs were fitted with new lyrics in Tày as part of party and government propaganda efforts. Papers in the conference volume conceded that Then rituals contained 'feudal superstition', but nevertheless described traditional Then rituals in meticulous detail and on their own terms.

One of the papers included in the 1978 conference volume was a report from what was then Bắc Thái province, where the Then are described as having 17 'songs', or stages in their ritual:[22]

1. Presentation of Offerings at the Beginning of Spring
2. Choosing Horses for the Voyage
3. Presentation to the Ancestors
4. Severe Prohibitions
5. Planting Out the Rice
6. Recruitment of People for Corvée
7. Presentation to the God of the Locality
8. The Hunt for Deer and Sambar Deer
9. The Ogress Da Dun
10. Putting the Rice into Storage
11. Crossing the Sea
12. The Market of the Three Luminaries
13. The Gate of Flowers
14. The Gate of the Master's Ghost
15. Invitation to the Generals and Inspection of the Offerings
16. Call to the Spirits of the Terrestrial World
17. Feast for the Soldiers

Many similar stages are found in the Pụt Tày ritual from Ba Bể district in Bắc Kạn, which has a total length of 3030 lines:

| | | |
|---|---|---|
| 1. | Invitations to the Master-Pụt | lines 1–61 (61 lines) |
| 2. | Inspection of the Offerings | lines 62–80 (19 lines) |
| 3. | Invitations to the Ancestors | lines 81–259 (179 lines) |
| 4. | Presenting Offerings to the God of the Locality | lines 260–469 (199 lines) |
| 5. | Passing through the Gate of the Mandarin-Genie | lines 470–492 (23 lines) |
| 6. | The Field of Ghosts of Undeserved Death | lines 493–537 (45 lines) |
| 7. | The Young People Choose their Company | lines 538–704 (167 lines) |
| 8. | Fight with the Lady Da Dun | lines 705–1047 (343 lines) |
| 9. | The Hunt for Deer and Sambar Deer | lines 1048–1132 (85 lines) |
| 10. | The Ape's Mother and the Field of Cicadas | lines 1133–1165 (33 lines) |
| 11. | The Ba Soi Leaf and Field of Graves | lines 1166–1261 (96 lines) |
| 12. | Giving the Orders | lines1262–1368 (107 lines) |
| 13. | The Banyan and the Beauty in the Moon | lines 1369–1412 (42 lines) |
| 14. | The Place of Vinh | lines 1413–1577 (165 lines) |
| 15. | Entering the Gate of the Paternal Ancestors | lines 1578–1642 (65 lines) |
| 16. | Entering the Gate of the Paternal Ancestor | lines 1643–1780 (138 lines) |
| 17. | The Market of the Three Luminaries | lines 1781–1980 (200 lines) |
| 18. | Passing by the Gate of Lucine | lines 1981–2000 (20 lines) |
| 19. | The Field of the Immortal | lines 2001–2027 (27 lines) |
| 20. | The Field Where One Chooses Flowers | lines 2028–2050 (23 lines) |
| 21. | Crossing the Seas | lines 2051–2714 (664 lines) |
| 22. | The Country of the Sky | lines 2715–2831 (115 lines) |
| 23. | The House of the Master-Sorcerer | lines 2832–3030 (199 lines) |

Overall, these stages are all part of a journey, starting in the place where the ritual is being held, whether in the main hall of a domestic dwelling or outside in the village, and leading up to the palaces of the highest deities in the heavenly realms. The words sung or chanted by the Then and Pụt take the form of what seems to be a narrative, describing the

places visited along the way, the necessary ritual actions, and the immortals, genies, and demons met at each stage in the journey. Unlike narrative in stories and novels in verse, where the action is conventionally understood as taking place in the past, the process of the journey in these rituals is understood as taking place in the present, and with the action ongoing as it is sung. Moreover, in true Vedic-Brahmanical fashion, the words are understood as bringing about the reality of what they describe (on which see further below).

## 5. Stages on the Road

In what follows, I will describe the stages in the journey to the sky, using the Pụt Tày text as an example.[23] Internal evidence indicates that the purpose of this particular ritual was searching for the lost vital spirits of a sick person, making appropriate offerings to all relevant deities, and bringing the vital spirits back and reinstalling them in the human body. In this section, the passages quoted have been chosen, first of all, to demonstrate that the ritual text is in fact structured as a journey to the sky; and secondly, to show some of the ways in which the journey has been indigenised; and finally, to give some coverage to other parts of the text that are either 'extraneous' interludes or serve as invocations.

The first two of the above stages involve actions that are necessary before the shamanic retinue sets out and thus are preparatory in nature. In the first section, the Pụt briefly rehearses his own personal history as a Pụt, then issues an invitation to his living and departed masters to accompany him on the journey in order to protect him from demons. He also requests iron wire to use for punishing evil creatures and declares that he will be able to find the vital energies of the sick person and thus accomplish the aim of the ritual.

In the second song, the Pụt lists the offerings and other things that have been prepared. These include handfuls of paddy rice, two cockerels to call to the lost vital spirits, a seal, a bell, a chair and a table made of special wood, two bundles of betel nut, two bundles of fruit, silver money in a packet, rice for the horses, a bowl of uncooked rice to hold the incense sticks, and a bowl of rice and an egg to serve as a vehicle for the vital spirits once they have been located. Finally, there is a bag containing clothes belonging to all the family members. All these things need to be mentioned in order to record their existence and make them transferable to the spirit realm.

In the third section, the invitation to the ancestors of the house takes but a couple of lines, and then the ancestors are invited to show the retinue of the Pụt the door of the house. Then follows a description of the house ladder down which the retinue must descend, and other things seen in the immediate environs of the house:

> The king of the household with the helmet embroidered with flowers  115
>   can guide you,
> The king of the household with a cast-iron helmet can show you the path.
> He wakes up to show the path to the sorcerer's masters,
> He wakes up to guide us.
> While moving toward the exterior door, with his hand he draws some
>   talismans,
> Once he reaches the window, he draws with his hand a phoenix,  120
> Go to the stair with a black pine cone in order to wash the feet,
> Go down the wooden staircase of *nghien* wood.[24]
> If a stair is going to break, it can be propped up with the wood of
>   the *nghien*,
> If the staircase is going to break, it can be replaced with a tree trunk.
> It takes three days to make the staircase,  125
> It takes seven days to repair it.
> The track where one can go up and down,
> Transit from where companions of the same age as the patron
>   enter and exit has thus been accomplished.
> We go until the end of the village of the young girl,
> Until the end of the path of the beauty.  130

The Pụt retinue then continues to the edge of the village:

> The low fence halts the pink hen,
> The high fence halts the purple hen.                                             155
> The cockerel strikes his wings three times, heaven's doors open up,
> He strikes his legs three times, the sky lights up.
> The group arrives at the garden of flowers in front of the yamen,
> At the garden of flowers in front of the village.
> He arrives at the plum tree to pick up the donkey,                               160
> He arrives at the fig tree to pick up the horse.
> Children, release the boat, mount the boat to go to the sky.
> Children, release the horse, climb on the horse, put yourselves on
>     the way,
> Release the phoenix, climb onto the phoenix to continue on the path.
> Each of you jump on your horse, quickly like a cat.                              165
> The masters jump on the horse, just like the swallows.
> By riding on horses you are going to become mandarins.
> We go along the path thanks to supernatural power,
> Thanks to the talisman our horses run with a light step.

The army and horses pass by the banana gardens on their way uphill to the temple of the god of the locality:

> The army of heaven immediately rows the boats                                    225
>     and gets on the horses.
> So get up, those who bring the incense,
> The servants, the porters!
> Servants, carry the burden on your neck,
> Carry the yoke on your shoulders.
> Go quickly on the path to heaven.                                                230
> Go joyfully, in front of me, the Pụt, leading the way!
> The army crosses by the plum tree near the pond,
> It arrives at the banyan tree at the entrance of the village,

The fourth stage in the journey to the sky is the visit to the temple of the god of the locality, a temple usually located on the outskirts of the village or some distance away up a hill. Here the retinue halts, the porters put down their loads, and offerings are made to the temple god, along with an announcement that the assembled party is on its way up to make offerings to heavenly deities. The primary offering here is 100 betel quids, consumption of which by the temple god opens the border gates wider and wider. In Tày culture this chthonic god is responsible for opening the road to the sky, and this function is actually articulated in the words of the song. This done, the entourage sets out again on the upward journey:

> Come back, assistants in charge of the incense,
> Get up, servants and bearers.
> Bearers, put the burden near on your neck,                                       345
> Carry the yoke over your shoulder,
> Carry it quickly to the kingdom of heaven,
> Rise up, virgins in charge of the incense,
> Come back, virgins responsible for the lamps.                                    350
> Disciples with ten years of service, masters with nine years of service,
> We are going to leave all, disciples since ten years,
>     masters since nine years.
> Get up, all of you, the two horses with three saddles,
> The twelve phoenixes belonging to the heads of the army.

> After drinking some wine, we saddle the horses, 355
> After the lunch break, the masters get on the horses.
> In the front are pitched the red and black flags,
> In the back are pitched the green and pink flags.
> From both sides, the rifles are ready, we go.
> Soldiers aligned two by two are advancing.
> While waving the flags we move toward the front door, 360
> And the order is given to the horses to start up.
> We arrive at the foot of the plum tree with speckled leaves,
> At the top of the plum tree with curved leaves.
> Its foot will be used as a gutter which directs the water
>     to the square metre of land in order to produce the rice crop,
> Its trunk will be used to make a sheath for the sword drawn by 365
>     the dragon.
> Its leaves will be used to manufacture the veil
>     for reaching the country of the Hác.[25]
> We rush the horses to follow the small stone path,
> Up to the spring where mortals draw water for the food supply,
> Up to the trough of water where one can bathe.

Arriving at the spring, the group dives into the water and makes a subterranean detour via underground watercourses, riding fishes and dragons down to the realm of the king of the underworld. Then, having imprisoned a giant turtle, the army travels back upstream and returns to the higher hills at the edge of human habitation:

> The army arrives at the foot of the *kham*[26] tree to take some rest,
> To the place where the grass can hurt the feet,
> Where the straw can cut the feet, 430
> Where the mountains' stone can hurt the knees.
> The horses reach the clumped field that no one harrows,
> To the wetfield *Na Bua* that no one transplants,
> To the place where on three sides are slash-and-burn fields,
> To the rice paddy with three locations to let water flow out 435
> At the three slash-and-burn fields sowed with cotton plants,
> To the three locations sowed with mulberry trees,

The fifth stage involves passing through the field of the dead grasshopper, itself the child of a high-level mandarin, where the entourage are enjoined to avoid spitting or blowing their noses out of respect for the dead insect.

The sixth stage involves the entourage passing through three fields inhabited by the ghosts of those who died unnatural deaths, by accident or violence, and then arriving at a crossroads where a young girl is weaving cloth. They move on past married couples who are blind, near-sighted, deaf, or injured.

The seventh stage begins with the horses negotiating stones and trees, and then on to the place where marriages turn out to be unfortunate. There is a digression on men and women choosing spouses and lovers. A contrast is drawn between the behaviour of wives and lovers after a man dies, where the wife observes the conventional fasting after death, and the lover abstains from meat for only one meal. There follows a disquisition on lazy women and intelligent women, comparing the kinds of cloth that each is able to weave and the ways they treat their husbands. Another one follows on about foolish boys.

For the eighth stage the entourage goes up to the cavern where the ogress Da Dun is taking a nap. Da Dun is a dangerous and malevolent creature, described as follows:

> Da Dun sitting, in a majestic way, on the edge of the pond,
> Her two knees are beyond her ears,
> The moldy skin is like that of a snake,
> Small mushrooms grow on her forehead,                940
> Her broken nose is like that of a monkey on a tree,
> Her large teeth are like bananas, full of food traces,
> Her nostrils full of hairs,
> Her skinny face, the long tongue pendant,
> Her hair not brushed and combed, simply horrible.    945
> Her two protruding eyes, black in color, frightening,
> Her withdrawn upper lip shows the teeth,
> Her red tongue goes down below the chin.
> Anyone who sees her, starts shivering in fear.
> The food residues stick on her lips,                950
> The young Pụt starts moving forward in order to engage in a fight.

The Pụt must engage in a battle of magic with Da Dun and vanquish her, in order to make her hand over her magic baton, which is needed for clearing all supernatural obstacles on the road to the sky. He tells her he plans to give it back to her once the mission is accomplished and the lost vital spirits have been retrieved. The struggle goes through a number of stages before Da Dun finally admits defeat. Da Dun gives the Pụt the baton and the entourage continues on its journey, flags flying.

The ninth stage in the journey is one in which the Pụt needs to hunt in the forest for deer and sambar deer, in order to supplement the existing sacrificial offerings, 'a hundred species for the sacrifice'. The hunting party's departure is described:

> Immediately the great prince transmits the order given.
> The prince gives orders to the skilled and strong soldiers to advance,    1055
> The best-chosen soldiers with three hundred guns.[27]
> They hold in their hands some gun locks, some bows, and crossbows.
> Recruiting the skilled boys who can use the gun,
> Seven thousand guns set out, shaking the path.
> The army follows them in large number,                1060
> The heroic chiefs are both talented and skilled,
> The traps are put in place as they march,
> Surrounding the forest paths.
> Here they employ three hundred good dogs,
> Eighty male dogs with white fur.                     1065

There follows a passage in which a mother deer hears the approaching hunters and gives instructions to her children, warning them against the danger of eating rice and other crops that belong to human beings. As the hunting party draws near, the animals flee:

> They climb up and hide inside the Stone Mountain.
> The noise of the smashed trees, the broken ivy frighten them,    1105
> The rotten trees are cracking, the branches are breaking.
> One can encounter some roe deer, some stags, some monkeys,
>     some gibbons, some wild boars,
> Some foxes, some wild cats, some birds, some rats. All of them crying:
> Which way to take in order to run away to safety?

Finally, with some animals wounded by bullets and others caught in the traps, including tigers with wounded feet, the porters shoulder the new burdens and the party sets out again.

The 10th stage involves the entourage moving through the fields at the foot of the highest mountains, where there are fireflies, cicadas, and monkeys carrying their babies. The cicadas are significant because the spirits of young boys and girls are attracted by their singing, and hence may be found lurking nearby.

In the next stage, the 11th, the army moves up to the top of the mountain, to a place where ghosts and demons lurk:

> During this existence one has to suffer next to the souls of dead children.
> The group arrives at the place of the young ghosts,
> At the place of the old ghosts that have become demons. 1195
> The young ghosts are sitting down in the middle of the procession.
> The demons are sitting down on the edge of the forest.
> Their beards are thick like the grass in the middle of the abyss,
> The hairs of their nostrils are as thick as ropes.
> I am not afraid of the appearance of devils, 1200
> Nor the appearance of the ghosts of young people.
> Flee far away, avoid my presence,
> Take yourselves off, put yourselves at a distance.

Then follows a description of the fates of those who died unnatural deaths, including new-born infants, women who died in childbirth, those who died of prolonged illness, and those whose fate ordained that they remained childless. The entourage then arrives at the foot of the ladder up to the sky:

> Just now examine the presents to be offered to the sky.
> Examine the propriety of the men who carry them on the voyage, 1345
> Before arriving at the ladder with its three rungs for mounting to the sky.
> After five rungs one mounts to the sky so blue,
> This rung I take hold of in order to dance,
> For coming and for going.
> I will go to the seven-gated earthly world to give place to a sunny day, 1350
> With eight gates for giving place to a rainy day.

At the thirteenth stage the entourage arrives at the gates of the sky:

> We pass by the fields to raise the line of silk, 1380
> Before arriving at the place of the beauty of the moon.
> At midnight we arrive at the gates of the sky,
> At dawn we see the gates of the sky so high as to be lost to view.
> They are of the yellow colour of ivory.
> The floor of the sky has the colour of copper. 1385
> We find the altar of the sky amidst the sound of gongs and drums.
> The hens of the sky have no feathers,[28]
> The cockerels of the celestial realm have no claws.
> The cockerel of the sky crows at the hour of the rat,
> The cockerel of the human world sings at the hour of the tiger. 1390

The entourage moves on to Yangzhou and Longzhou, which are described as idyllic places:

> The young girls come here to draw the water.
> They return with a bucket of water and a bucket of fish.
> Men come here to gather the thatch, 1405
> They come back with a load of flowers and a load of thatch.
> In this ricefield the rice is white without being pounded,
> The vegetables are clean without being washed.
> The water flows through the ricefield without being irrigated.
> The fish leap up to look for the men. 1410
> The fertile earth is favourable for raising buffalo cows that will
>   give birth to lots of buffalo calves.
> The pigs grow rapidly.

The next stage in the journey is the place of Vinh. This is a long segment recounting a Tai version of the story of Dong Yong 董永, one of the 24 exemplars of filial piety in China. The narrative here is devoted to describing the filial devotion of Dong Yong toward his

mother, with a storyline that is similar to the Zhuang and Bouyei versions of this tale that are found widespread in Guangxi and Guizhou.[29] Its location here, in the middle reaches of a shamanic journey to the sky, may be related to the concern, mentioned previously in these lyrics, that the people worthy of presenting offerings to the highest deities should themselves be 'proper people', conforming to the expectations of their society, considerate of others, and respectful of their own parents.

The 15th stage describes a place in the sky where the mansion of the ancestors is located. Here are a few lines from the description:

> My children, have a look,
> Here is the house of the sky.
> We see the faces of our ancestors,
> The house of the mandarin in charge of our ancestors.                              1585
> Doves whiten the roof of the house,
> White cranes whiten the roof of tiles.
> They come to perch there to see the house of the mandarin
>      in charge of our ancestors.
> His two or three houses are covered with tiles,
> His four or five houses are covered in bronze.                                      1590
> The roof resembles the scales of a pangolin,
> Green moss pushes the four sides, the solemn columns are
>      painted red,
> The ends of the cornices are multi-coloured like a snake with white
>      bands.
> The roof is curved like a pair of horns,                                            1595
> The partition walls are speckled like the wings of an eagle.

And then:

> The army passes in front of the gate, in front of the house.
> Going up, one sees the ducks preening their feathers with their beaks.

And then there are difficulties finding the road forward:

> We arrive at the place of the princess.                                             1615
> The princess opens a fan decorated with flowers,
> The guardian of the sky opens a fan of rattan.
> Their sleeves sweep up the sun,
> The fan of silver goes out in front,
> He sweeps up the clouds which dissipate,                                            1620
> He sweeps up the dew which flows.
> When I part, the clouds dissipate,
> When I traverse, the dew evaporates.
> Here one finds the black night which blocks the way,
> Here, one finds the black coat which conceals,                                      1625
> I am not able to find the road.
> The ancestors light the lamps to help me find the road,
> The masters light the big lamps to to indicate the way to me.

And then on to an enlivened landscape:

> I arrive at the place where the karst peaks come to salute,
> Where the grand mountains encircle me.                                              1630
> I arrive at the mountain where nine gorges prostrate themselves.
> I arrive at the flat rock,
> At the elevated outcrop.

> I ascend, and the mountain leads me up.
> I arrive at the golden rice with its broken leaves,      1635
> The buffaloes of gold and silver eat these leaves.
> I arrive at the place of waterfalls of noisy water,
> At the place of Kwan Yin and of old sorcerers.
> I arrive at the place of the founder of the sorcerers.

The 16th stage is one with vast open wetfields with no rice planted in them. Crossing over a copper bridge, the army and horses then arrive at the street of the Tam Quang market.

> Then we go up to the source of the golden water of the sky.
> We traverse the source of the silver water in the sky,      1675
> To arrive at the foot of the pine tree with its pendant leaves.

And then on further upwards:

> I come to the place at the foot of the areca palms,
> And I arrive at the road where the dead and the newly-born are
>     registered.
> The god of the North Star is seated on the right,
> The god of the Southern Cross is seated on the left.
> Each day three thousand people descend to the earth,      1695
> Five hundred return to the sky.

and upwards (lines 1701 ff.):

> This night I ascend to the gate of the sky,
> Tomorrow morning I will ascend to the gate of thunder.
> The thunder strikes the tree in the middle of the bush,
> It splits the trees in the middle of the forest.

And then the retinue approaches the sea up in the sky and arrives at the place of the widows.

The next stage is the market of the Tam Quang, the Three Luminaries. The Three Luminaries (Ch. Sanguang 三光) are identified as the Sun, Moon, and Stars. The name indicates that at this stage in their journey, the Pụt retinue are already up in the sky. Pigs need to be purchased here in order to ensure that the offerings to the highest deities are complete in every respect. The Pụt delegates two young acolytes to go into the market, find someone there who sells pigs, buy them, and bring them back. At the same time, other boys are sent to the market to purchase bowls and platters. On their way into the market the boys encounter two girls, who tell them that their father sells pigs, and they invite the boys to follow them back to their father's house. Catching sight of them, their father immediately assumes that the boys have come to seek his daughters' hands in marriage. Disabused of this impression, their father agrees to sell the pigs, and the transaction, including the weighing of the pigs and haggling over the price, is eventually successful.

The Sanguang are a prominent feature of the Daoist school of Tianxin zhengfa 天心正法, 'Orthodox Methods of the Heart of Heaven', one of the Daoist schools that emerged during the Song dynasty (960–1279 CE) (see Li 2011). Here, however, the focus is exclusively on the market transaction.

Setting out from there, the entourage arrives at the great crossroads in the sky:

> I go across three hundred great crossroads,
> Nine hundred little crossroads,
> From which the roads of the world set out.
> One route goes to the country of the Nông who work in the rice-fields.

> Another goes to the country of the Hác who work in the dry-fields, 1965
> One road goes to the country where Chiêu Quân was sent,
> One road for going on a tribute mission to the country of the Ngô.
> One road conducts one to the God of Jade,
> Another to the terrestrial world to become a master-sorcerer.
> Another to the watery world of the Dragon King, 1970
> One route for going on a tribute mission to the market of the
>    golden buffalo.
> One route goes to the country of the Lucine,
> One route goes as far as the country of the sky,
> One route goes to the tribunal in the sky.
> I have found three hundred great crossroads, 1975
> I have gone through three hundred great crossroads,
> Nine hundred little crossroads,
> The routes meet in three hundred and twenty places.
> I am the road to the immortal lady,
> I choose the road for presenting the offerings to the Mother.

The Nông and Hác are mentioned frequently in these lyrics, and refer respectively to the Nùng people and Han Chinese immigrants (the name Hác means 'guest' 客 *ke*).[30] Chiêu Quân in line 1966 refers to Zhaojun 昭君, and Ngô is the Vietnamese transliteration for Wu 吳, the name of a kingdom in the south of China in the third century during the Three Kingdoms period.

The 18th stage contains mainly the lyrics of a song singing of the activities proper to each month in the year, including the cultivation and harvesting of rice. The significance of this song is connected with the importance of rice, and especially uncooked rice in a bowl along with an egg, which serves as the vehicle by which the vital spirits, once found and rescued, can be transported back down to earth.

The 19th stage, the field of the immortal lady, indicates another form of lurking danger. The immortal beauties in the sky are all eagerly looking for husbands, and the Pụt, in order to accomplish the goal of the mission and return safely, must spurn their advances.

> This place has no officials, 2005
> The immortal lady has no husband.
> Each day she goes forward along the road to marry herself to an official.
> Each day she goes forward along the road to find a husband.
> She keeps on rolling up leaves of betel to chew,
> To cut the areca nuts. 2010
> The betel quids are carried in her mouth,
> The cut-up areca nuts fill the little plates,
> But as a Pụt, I have my own quids of betel,
> I have brought along my own areca nuts.
> They are always in my pocket, 2015
> While marching along I eat them,
> When marching along I chew them.
> My quids of betel are not lacking for anything,
> I have no need to look for any others.

And further:

> Yesterday my wife received me at the house, 2020
> She had returned the day before.
> Her skin is of a tender whiteness,
> She is more beautiful than you.
> Ah well! Take yourself out of my road,
> Leave the road free and wide. 2025
> My soldiers occupy the route,
> I am going to look for the souls, the vital spirits.

The next stage, the 20th, is the garden of flowers. The garden of flowers is connected with the conception and birth of children, superintended by the Goddess of Flowers whose cult is widespread in southern China and northern Vietnam (see Cauquelin 1996). Each child has a floral counterpart in the garden in the sky, with golden flowers for boys and silver flowers for girls. Here the lyrics draw upon those of a well-known song, asking which flower blooms in which month, and then answering this question for the 12 months of the year. This theme also is related to ensuring that the lost vital spirits are alerted and allow themselves to be located and taken back down to earth.

The next stage is Crossing the Seas, and the Pụt's retinue must take to the boats, propelled across the Milky Way by the rowers:

| | |
|---|---:|
| They chew the betel and spit out the quids chewed up all red. | 2075 |
| The Pụt in the army make arrangements | |
| To write the orders to the Milky Way. | |
| The order is to row across the sea: | |
| It is a matter of rowing to the South in order to cross the Milky Way. | |

The fleet is of considerable size:

| | |
|---|---:|
| Fifty golden boats descend, | 2150 |
| Fifty silver boats arrive. | |

and:

| | |
|---|---:|
| The boats and rafts should be complete and ready. | |
| Here is the golden boat to transport the treasure, | |
| The precious objects that belong to the king in the sky. | |
| Here are the boats full of dignitaries and beautiful women. | |
| Here are the boats carrying young ladies who are also mandarins. | 2165 |
| The boats come down and fill up the crossroads. | |
| One boat transports the flowers from foreign lands, | |
| One boat transports the grilled rice, incense and flowers. | |
| One boat holds beautiful objects made of agate. | |
| One boat holds the rice and the wet-rice of the household. | 2170 |
| One boat holds all the beautiful women. | |
| One boat holds the swallows and golden orioles, | |
| One boat for the incense and the flowers, | |
| One boat for the swords, the spears, the shields, | |
| One boat for the carriages, the donkeys, the horses, | 2175 |
| One boat for the draperies with their phoenix patterns, | |
| One boat for the young immortal maidens, | |
| One boat for the boys and youth, | |
| One boat for the queen and the royal concubines, | |
| One boat full of objects belonging to the great Buddha, | 2180 |
| One boat for the incense and the flowers for the old Buddha, | |
| One boat for the ancestral mothers of the land, | |
| One boat for the Buddha with elevated powers, which comes and goes, | |
| One boat for the adopted children of the Pụt. | |
| One boat for the adoptive children of the ladies. | 2185 |
| One boat for the soldiers, | |
| One boat for the generals, the sons-in-law of the king. | |
| Everywhere the soldiers prepare the boats, | |
| Then come the kings who share out the horses. | |

The itinerary takes the entourage, propelled by the rowers, across the Milky Way to the continent in the sky where the world's highest mountain is located. The actual journey in the Then narrative is described in terms of upstream and downstream movement, along the lines of other migration narratives of Tai-speaking peoples (see Holm 2009), rather than a journey from asterism to asterism. So the boats are described as going upstream and then

downstream through a series of rapids. These are numbered but not named. The end point of the journey, however, is clearly stated as Mount Sumeru (phya Su Mi).

Once on the other side of the celestial seas, the Pụt entourage arrives at the next stage, the Country in the Sky. Here the narrative turns to recounting the old cosmogonic myth, widespread in southern China and Southeast Asia, of how the sky in primaeval times was very low, and of how there were then 10 suns, which caused years of drought and famine until they were dispatched; then of how the Thunder God sent incessant rain, leading to worldwide flooding; and of how a brother and sister pair took refuge in a gourd and became the last survivors of the human race; and of how they subsequently became man and wife and gave birth to a monstrosity, which they then hacked to pieces and threw into the forest; and finally, how the pieces came to life and became the forbears of various ethnic groups. The narrative then goes on to recount the origin of family quarrels over inheritance and other things, leading to generations of enmity.

The entourage of the Pụt then moves onward for the final stage in the journey, through a series of fortified gates leading to the palace of the highest celestial gods, where they will be received in audience:

> The sound of the bell goes from the sky to the world of the humans.
> People fetch water for washing their feet.
> The sky has the multi-coloured aspect of a nest of bees.     2845
> The clouds wind up like a nest of bees.
> The Pụt are seated on the hard spines in serried ranks.
> It is indeed rare that one can see the whole celestial country.
> It is indeed rare that one can look out on the whole country of
>     the immortals.
> The news has arrived at the first gates.     2850

And:

> One calls to the master to open the gate of the sky,
> One demands of the master to open the great gates wide.
> One opens the gates with their pointed nails,     2855
> The gates of gold and silver are open in great numbers.
> The news has arrived at the second gates.
> One enters the garden of mulberries,
> On hangs up the threads of silk mixed with grains of paddy.
> One arrives at the owls who guard the gates,     2860
> The beauty who guards the incense demands that the sky
> Make the announcement to the third gate.
> The great gates are covered with cast iron,
> The public gates carry the imprint of seals.
> The gates of the generals are protected,     2865
> The gates of the master are protected by the Buddhist divinities.

And then finally, a description of the celestial audience:

> Girls and boys stand upright, solemn on both sides.     2945
> I walk, imposing in the middle.
> The Pụt gather together in front of the table,
> Under the parasols and the umbrellas like mandarins of the state.
> Watching the Pụt come out together.
> The beauties are standing together upright,     2950
> The young girls and young boys, in serried ranks
> Inviting the Pụt and the mandarins to take their seats.
> I the Pụt am present, the tall hat on my head,
> The robe embroidered with dragons in my hand.
> One puts the hat on the head of the great mandarin who governs     2955
>     the land,

> One awards a certificate to the great mandarin who administers
>   the villages,
> The Pụt of the sky allow me to enjoy the goods in perpetuity.
>   All the world knows our teachings.

In this edition the actual petition, the presentation of offerings, the securing of the vital spirits of the sick, and the return journey are not included. In other texts they certainly are, along with descriptions of the celebratory feasting for the mandarins, porters, rowers, soldiers, and horses once the entourage has arrived back on earth. Giving appropriate thanks to all members of the party are particularly necessary: the lyrics of the rowers crossing the seas make mention of the hardships endured by the rowers' wives and children while they are absent on their assigned duty in the sky.[31]

This overview of the stages in the journey to the sky, and the selected passages quoted, demonstrate clearly that this Pụt ritual is an archetypical shamanic ritual. The Pụt and accompanying entourage make their way up to the sky in person, rather than sending spirit emissaries, encountering any number of different obstacles, crossing the celestial seas, and finally reaching their destination and presenting all the offerings necessary for the successful accomplishment of their mission. Like the native chieftain in the native chieftaincies of the China–Vietnam borderlands, who leads his or her troops into battle personally on military campaigns, rather than assigning the task to a subordinate, the Pụt or Then leads his or her entourage in person (Holm and Meng 2021, p. 33). The entourage itself is variously described at different stages in the journey, and consists of soldiers, horses, generals, porters, servant girls, rowers, all accompanied and protected by the spirits of the Pụt's master-teachers living and dead, and bearing gifts including an array of sacrificial animals as stipulated by custom. A relatively full list of members of this travelling party is given when describing the loading of the boats prior to crossing the seas (as quoted above). It might be thought on one level that there is an element of hyperbole in these descriptions, but what is actually operative is a kind of Brahmanical power of words, whereby saying something has the effect of making it real.

The entourage is also likened to a tribute mission, led by the Pụt as ambassador to the celestial court in the sky. Tribute missions to the Chinese imperial court, bearing a variety of gifts including local specialities, were a salient and essential part of international relations for the Vietnamese court right up until the end of the imperial period (see Nguyễn Thị Kiều Trang 2016). There are in fact other Pụt and Then texts called *Pây sử* 'Going on a Tribute Mission', so this parallel is one that is overtly recognised (see e.g., *Tổng tập truyện thơ nôm các dân tộc thiểu số Việt Nam* 2015). For that matter, it was reported at the 1975 conference on Then that everybody in Tày and Nùng communities was aware that Then rituals were journeys up into the sky (*Mấy vấn đề về Then Việt Bắc* 1978, p. 94). Moreover, at the same conference it was reported that some of the songs sung in Then rituals were modelled on the songs that had been sung by tribute missions sent by the Mạc court in Cao Bằng to the Chinese capital (*Mấy vấn đề về Then Việt Bắc* 1978, pp. 144, 146).

The shamanic journey up to the sky is based on Vedic rites of ascension (*vājapeya*), which are mentioned in the Vedas.[32] Detailed instructions for this ritual are given in the *Śatapatha Brāhmaṇa*, including instructions for felling the tree out of which the required sacrificial post is made.[33] This rite of ascension is widespread in many areas of southwest China and mainland Southeast Asia, as well as in Central Asia, where the ability of shamanic practitioners to ascend to the sky frequently takes the form of climbing a ladder of swords.[34] The ability to climb a ladder of swords without injuring oneself is listed as one of the attributes of the Buddha in the earliest Buddhist treatise in Chinese, the *Mouzi* 牟子, a text which dates from the late second or early third century CE and has links with early Buddhist centres of learning in the Red River Valley (Holm 2018, pp. 21–22; citing L'Haridon 2017, pp. 11–12).

## 6. Indigenised and Indic Elements

As previously indicated, there are many stages on this journey to the sky which now relate to local or indigenous customs, and others which reflect Chinese cultural elements. Thus, for example, in the imperial palace in the highest heavens we find the Jade Emperor Ngọc Hoàng (Ch. Yuhuang dadi 玉皇大帝). Many of the other figures mentioned are Chinese, such as the exemplar of filial piety Dong Yong. But the Jade Emperor is often mentioned in parallel with other figures such as the Big Buddha, the Old Buddha, or Mother Buddha. This last also appears in these Tày texts as Mẻ Tsik-ka 'Mother Śākyamuni', with the name 'Śākyamuni' represented by the transcription of just its first two syllables; the first syllable of this designation, Mẻ, is the Tày word for 'mother' but also a noun head for the names of mature-age women. These figures are mentioned along with the Jade Emperor as recipients of offerings in the highest heavens.

There are two elements, however, which point unmistakably to an underlying Indic framework. These are the name of the highest mountain in the sky, Sumeru, and the oceans in the sky which the shamanic retinue must cross in order to reach their destination. We will look at these next.

## 7. Sumeru and the Heavenly Seas

In the ordination ritual of the Then, the world mountain, phya Rú Mi in Tày transcription, is mentioned frequently, as well as serving as the destination of the celestial journey (Nguyễn Thị Yên 2007, pp. 374, 381, 394, 395, 448, 459, 465, and 525). In the 'Ritual to Remove Astral Obstructions' (Giải Hạn) text, there is an entire section devoted to the Then retinue's ascent of this mountain.[35]

Sumeru as the highest mountain in the world and the abode of heavenly deities is first mentioned in the epic *Mahābhārata*, and belongs to a wider Indian worldview, rather than being confined to Buddhism.[36] Still, in canonical sources on Buddhist cosmology Sumeru is the central world mountain, and is described as surrounded by concentric seas and mountain ranges, which separate Sumeru itself from the continent inhabited by human beings, Jambudvīpa. There were altogether seven mountain ranges surrounding Sumeru, with seas separating each of these mountain ranges, and finally the four continents including Jambudvīpa ranged around the outside.[37] Thus, in order to reach the base of Mount Sumeru and begin the ascent to the realm of the higher gods, our shamanic retinue from the world of mortals would have to cross not one, but eight seas, as shown diagrammatically in Figure 1 below.

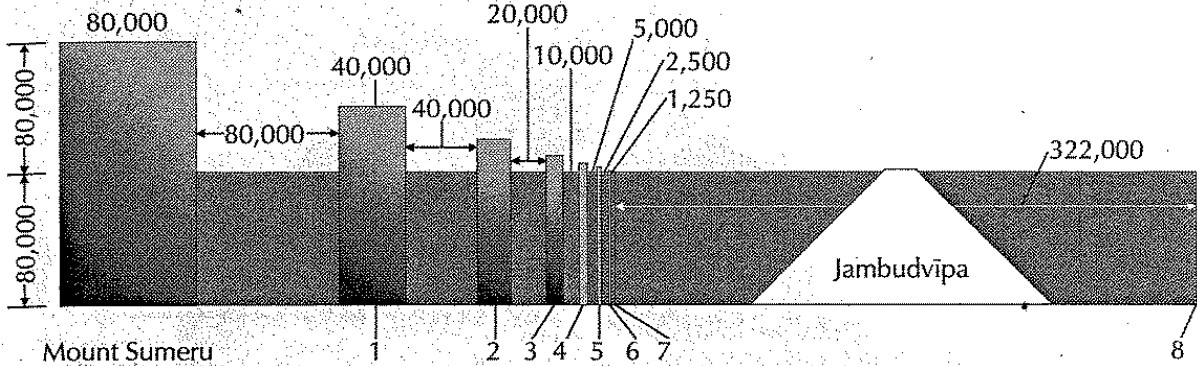

**Figure 1.** Sizes of Mount Sumeru and Surrounding Mountains and Seas. (Source: Sadakata Akira 1997, Fig. 7) (1) Yugandhara (40,000); (2) Īṣādhāra (20,000); (3) Khadiraka (10,000); (4) Sudarśana (5000); (5) Aśvakarṇa (2500); (6) Vinataka (1250); (7) Nimindhara (625); (8) Cakravāḍa (312.5). (Dimensions given in *yojanas*.).

It is this circumstance that explains the ubiquitous presence of "Crossing the Sea" segments in the rituals of the Then and Pụt in northern Vietnam. Moreover, it seems that

specific episodes in the narrative of crossing the seas in these ritual texts correspond to specific features of the canonical description in the *Abhidarmakośa*: the encounter with the circle of wind, the colors of the waters, and the description of Sumeru itself.[38]

The *Abhidarmakośa*, a voluminous treatise with commentaries, was first brought to China and translated by the Indian monk Paramārtha (Zhendi 真諦) sometime in the first half of the sixth century CE, with the title *Apidamo jushe lun* 阿毗達磨俱舍論. Knowledge of its contents may have circulated earlier, but the point is that it was available in Chinese from a relatively early date. For Sumeru, the transcription Su-mi-lu 蘇迷盧 is found in this Chinese translation (see *Yiqie jing yinyi* 2010, ch. 70). The name of this mountain in the texts of the Then and Pụt is 首眉 (Ch. Shǒu Méi), transcribed variously in Tày as Xu Mi, Su Mi, or Rú Mi. We can be confident in identifying this mountain as Mount Sumeru. In Chinese Buddhist texts Sumeru is given various renderings, the earliest being 須彌 (Xu-mi), but 須彌樓 (Xu-mi-lou), 修迷樓 (Xiu-mi-lou), and 修迷留 (Xiu-mi-liu) are also found, as well as 蘇迷盧 (Su-mi-lu). The existence of two-syllable renderings in Chinese such as 須彌 gives us a close counterpart to the rendering in Tày.

Further afield, various forms of this name are found over a very wide area of southern China among Tai-speaking communities and indigenous ritual practitioners, often heavily indigenised and not recognised locally as part of an Indic substratum. In the Zhuang-speaking Tianyang area of central-western Guangxi, the name Couhmiz 州眉 is mentioned in connection with a narrative about the primordial flood, and Couhmiz is understood to be the name of a high mountain. In a Bouyei version of the flood myth from Wangmo county in southern Guizhou, the name Cojmiz (written 索密) is explained as the name of an old man who lives on the top of Bolangshan 播朗山, the only peak high enough to protrude above the floodwaters during the great flood. Even further north, in Zhenning in west-central Guizhou, a near-homophonous name (站走煤, tɕiaŋ²⁴ ɬeu⁵³ mei¹¹) is mentioned in funeral texts as the primordial homeland of the Bouyei people. In these cases, the name in disguised form has been reattached to versions of the flood myth, as the name of a place protruding above the floodwaters. What this indicates, however, is that the under-layer of Buddhist-Brahmanical elements was very pervasive in the south of China, as elsewhere.[39]

## 8. Brahmanism and Ritual Templates

All these Indic ritual elements are connected historically with the development of the early Buddhist centre of Luy Lâu, in the Red River Valley. Located to the east of present-day Hanoi, Luy Lâu was the seat of government of the Chinese-ruled commandery of Jiaozhi 交趾 during the Han dynasty (see Trần Văn Giáp 1932, p. 209). Inland waterways connected it with the mouth of the Red River and the open sea. By the early centuries CE, it came to rival the northern Chinese capital Luoyang as a centre of Buddhist activity in size, including the number of its temples and the number of monks. As a port of call for Buddhists and other priests from India, it also developed into an important centre for the translation of Buddhist texts into Chinese. Historical records indicate that there were many monks from India resident there, and over time it also became an important point of departure for Chinese Buddhists travelling to India.

Contemporaneous descriptions indicate that the variety of Buddhism introduced in those early days was not of the rarified kind that later developed into Chinese gentry Buddhism, but rather an amalgam that encompassed many popular elements drawn from disparate Indian traditions. Thus, the 'Treatise on the Resolution of Doubts' (*Li huo lun* 理惑論) of Mouzi 牟子, a text which can now be accepted as early and genuine (L'Haridon 2017; cited in Holm 2018, pp. 21–22), describes the Buddha as the enlightened one who can change shape at will, can climb a ladder of swords without being cut, and can walk on fire without getting burnt.[40]

Climbing a ladder of swords and walking on fire are just two of the South Asian ritual segments that are practiced widely throughout East Asia, Southeast Asia, and indeed also Central Asia, a region that was also subject to Indian influence. The specific protocols attached to these ritual procedures are remarkably uniform throughout this wide area.[41]

They have been incorporated into the rituals of religious practitioners who are not themselves Buddhist: they form part of Daoist initiations in Fujian province and elsewhere in China, and they are an essential element in shamanic induction ceremonies among the Tungusic-speaking Xibo people in Chinese Central Asia (present-day Xinjiang). There are other South Asian rituals and ritual segments that are similarly widespread: buffalo sacrifice, horse sacrifice, voyages by boat, and so on, with ritual procedures that are also remarkably similar, even though in most cases they are no longer recognised locally as coming from India.

The protocols, and detailed instructions for the performance of such rituals, are to be found in the Brāhmaṇas, written ritual compendia going back to the early post-Vedic stage of Sanskrit literature.[42] The Brāhmaṇas not only provided explicit instructions for the performance of rituals, but also provided detailed explanations for why the requisite ritual actions were prescribed in such a way. They tended then to be quite voluminous, leaving nothing important out of the account. Not only this, but they came along with a fundamental assumption that it was the words of the ritual, as performed correctly by Brahmanical priests, that produced the real results in the external world. It was the Brahmans' ritual performance that made the sun rise in the morning.[43]

Evidence indicates that Brahmans as well as Buddhists were among the earliest visitors to the Red River valley.[44] Their presence elsewhere in maritime Southeast Asia and mainland Southeast Asia, often in very considerable numbers, is well documented.[45]

### 9. Maritime Transmission

Buddhism and Brahmanism in the Red River valley, in Luy Lâu and other places, came via the maritime route, and not via Central Asia as has sometimes been suggested. A maritime connection between South Asia and the East Asian region, either direct or indirect, was already established by the beginning of the CE era.[46] The dynastic history of the Han dynasty mentions the 'barbarian ships' that were docking in Nanhai 南海 (present-day Guangzhou). The beginnings of sea voyaging and maritime commerce are now reasonably well known and began some centuries before the CE era with products and people carried in Austronesian ships. Later on, the Indians developed the technology for building their own oceangoing ships, basing their designs at first on Austronesian models. By contrast, it was only some centuries later that the Chinese developed the capacity for ocean voyages.[47]

### 10. Shamans and Spirit Mediums

Rituals structured as journeys to the sky are widespread in maritime Southeast Asia and mainland Southeast Asia, as well as in archetypically shamanic regions in Central Asia. This is well-established. One of the best-documented examples of a shamanic ritual in maritime Southeast Asia is to be found in Clifford Sather's *Seeds of Play, Words of Power*, which includes an interlinear text and translation of a ritual to locate and bring back the lost souls of a sick family member, along with copious ethnographic notes (Sather 2001).[48] Like the case discussed in the present article, the ritual is structured as a journey, and involves the shaman (*manang*) in singing or chanting through 'an ordered sequence of named places' (*ibid*., p. 173), from a starting point in the Iban longhouse, and then returning to the longhouse at the end of the ritual.

Given the postulated prehistoric connection between the Austronesian peoples and Tai-Kadai speakers, it should not be surprising that one can discern many common points of detail in these rituals, such as the use of chickens to locate the lost spirit entities, the pervasive use of semantic parallelism in liturgical song verses, the botanical metaphors used to refer to kinship relations, and so on. Similarly, there is much evidence of indigenisation of cultural categories in the Iban text: the highest mountain has been re-identified as Mount Rabung, a sacred mountain in the invisible world but also the name of the highest mountain in the Iban world, located in the Sri Aman Division in Kalimantan Barat. (*ibid*., pp. 116–17; for chickens, see p. 195).

## 11. Conclusions

The stages on the road to the sky, as performed by the Pụt and Then, are as we have seen indigenised to a considerable degree. However, the ritual framework itself, based on a Vedic rite of ascension, and certain stages in this journey, notably the crossing of the celestial seas and the designation of Sumeru as the central world mountain and final destination, are definitely Indic, even though they may not be recognised as such by local people. Direct Indian influence in East Asia seems to have declined by the end of the Tang dynasty (618–907 CE) in China, in other words around the end of the ninth century, and many of the ritual elements would subsequently have been overlain with layers of indigenised and locally appropriate interpretation, even if the ritual procedures themselves were preserved more or less intact. This process of localisation is very common throughout the East Asian region and can be seen as part of a process of recentering, whereby local communities lay claim to their important part in the history of civilisation. Within Buddhist institutions, of course, the dynamic would have been different, and memory of South Asian origins would have to have been preserved.

One of the main contributions of the present article is to present more detailed evidence of the actual song lyrics which carry the entourage of the Pụt or Then upwards through the various stages in their journey to the sky. These provide additional confirmation that these rituals are archetypically shamanic in nature and are structured as journeys up to the sky. As we have seen, these lyrics vary in their internal coherence as ongoing narrative, with digressions and insertions of extraneous material that need to be understood in terms of their underlying and unspoken ritual function. At times they are quite clear and uncannily beautiful. Many seeming digressions can also be understood as invocations, whereby mentioning things, people, or ideas has the effect of calling them up and facilitating their transfer to the spiritual domain and the shamanic entourage.

The task of explicating all these complexities has only just begun.

**Funding:** Research on which this chapter was based was supported by an Australian Research Council-funded large grant project on The Old Zhuang Script, 1996–1999. Subsequent support came from a research grant from the Science Council of Taiwan (MOST) from 2011–2017, project title 'Vernacular Character Writing Systems among the Tai-speaking Peoples of Southwest China and Northern Vietnam' (grant nos. 102-2410-H-004-055−, 102-2410-H-004-111−, and 104-2410-H-004-162−), and the National Science and Technology Council from 2024 to 2026, project title 'Mapping the Vernacular Tay Script' (grant no. NSTC 113 - 2410 - H - 004 - 093 - MY2). The publication of the special issue in which this article is included has been partly supported by the French National Research Agency (ANR), project MANTRA—Maritime Asian Networks of Buddhist Tantra (ANR-22-CE27-015), coordinated by Andrea Acri (EPHE, PSL University, Paris).

**Institutional Review Board Statement:** Not applicable.

**Informed Consent Statement:** Informed consent was obtained from all subjects involved in the research on which this article is based.

**Data Availability Statement:** Evidence on which this article was based is available in the form of published works. See especially Bế Viết Đằng and Lục Văn Pảo (1992), listed in the References section.

**Acknowledgments:** This manuscript was published in the special issue "Beyond the 'Spice Routes': Indic and Sinitic Religions across the Asian Maritime Realm," with Dr. Andrea Acri and Dr. Francesco Bianchini serving as guest editors.

**Conflicts of Interest:** The author declares no conflict of interest.

## Notes

[1] For a classic study see Durand (1959).

[2] Scholars writing about these rituals do not always make this distinction clear enough, referring for instance to Tày Then and Pụt as 'spirit mediums' (see e.g., Đoan Thi Tuyen 2012). All too often spirit medium practices as well are referred to as 'shamanic'. On such confusions see further below.

3    Tantric or Vajrayāna Buddhist practices developed in mediaeval India from the 5th–6th century CE and in China from the first half of the 7th century onwards. See Orzech (2011). On Tantrism see also Acri (2017).

4    See Strickmann (1996, p. 425). He was referring primarily to Shirokogoroff (1935, p. 269). For a more recent and meticulous discussion of this issue, see Slouber (2023, pp. 61–64).

5    The text is said to have been translated in the 1st century CE. On which see Zürcher (2007, pp. 29–30). Zürcher notes (p. 29) that the term *śramaṇa* appears in the 'Xijing fu' 西京賦 'Poetical description of the Western Capital' of Zhang Heng 張衡, a work completed at the turn of the century 100 CE, and that this indicates that the term was probably already familiar to the 'general cultured public' at the capital. For an example in the text see *Fo shuo* 2a.

6    See Eliade (1964, pp. 442, 442–44, 455). See also p. 269, on the shaman climbing the Cosmic Mountain at the Center of the World: "It is only the shaman and the heroes *who actually scale* the Cosmic Mountain, just as it is primarily the shaman who, climbing his ritual tree, is really climbing a World Tree and thus reaches the summit of the universe, in the highest sky." (italics in the original).

7    On Celestial Ascents in Central and Northern Asia, see Eliade (1964, 181 ff); on the World Mountain, also called the Cosmic Mountain, pp. 266–69.

8    Including folklorists Lauri Honko and Å. Hultkranz, mentioned in Holm (2018, p. 4).

9    Referring to southern China, mainland and maritime Southeast Asia, and Central and North Asia, he noted "both [regions] have seen their religious traditions definitely modified by the radiation of higher cultures" (Eliade 1964, p. 279). He declined, however, to give an "historico-cultural" explanation (same page). Cited in Holm (2018, p. 44).

10    'The 2009 Vietnam Population and Housing Census: Completed Results', General Statistics Office of Vietnam: Central Population and Housing Census Steering Committee, June 2010, p. 134.

11    A substantial proportion of the Tày population in the north dates back very nearly to the time of the earliest human habitation in this area (see *Các Dân tộc ở Bắc Kạn* 2003, pp. 77–78). This judgment is based on archaeological evidence but is also in line with DNA evidence, on which see Li (2002).

12    Lung Hin in Taiping fu 太平府 is located in present-day Tiandeng 天等 county, Long Châu is present-day Longzhou 龙州.

13    See Bế Viết Đằng and Lục Văn Pào (1992, pp. 166–67). Daoist priests are also found in northern Vietnam. It is significant that the Tạo (*daogong*) recite their texts in Southwestern Mandarin, even well to the south of the China-Vietnam border (Holm fieldwork, Chợ Đồn district, Bắc Kạn province, August 2015; Lạng Sơn province, August 2017).

14    The designation Pụt comes from an older pronunciation of the word 佛 *fó* 'Buddha', the Late Han and Middle Chinese pronunciation of which was /but/.

15    D. Holm, fieldwork, Jinlongdong, February 2016.

16    In 'Literate Shamanism' I go on to comment that the performance style of men and women is nevertheless quite different. During actual performance men can be seen reading from their copies of the liturgical manuscript, whereas the women recite the words of the same lyrics from memory. See Holm (2019, p. 6).

17    The texts are not presented in interlinear format and do not come complete with word glosses for the Tày words. For Tày scholars this may not present any difficulty, but the task of morpheme identification is required for international purposes.

18    See Bế Viết Đằng and Lục Văn Pào (1992, p. 171). Differences in handwriting are clearly visible in the photo-reproduction of the manuscript pages at the back of the volume, making it fairly easy to disentangle the various manuscript sources.

19    Typically these days the grounds of a school in the village.

20    For Tày and Nùng comparisons see Holm (2020b); for voiced initials see Holm (2023) and Holm (2024b). I have also written a book chapter on Then performance as an Intangible Cultural Heritage item in Guangxi (Holm 2020a), and prepared material for a critical edition of the Cao Bằng Wedding Songs collected by Nguyễn-văn-Huyên (1941); see Holm (2024a).

21    The stages and results in this very early survey of Then practitioners in northern provinces are discussed in Nông Văn Hoàn's (1978) introduction to the conference volume, 'Bước đầu nghiên cứu về then Việt Bắc', pp. 7–14.

22    See Triệu Đường (1978, p. 62). The province of Bắc Thái is now divided into two provinces, Bắc Kạn and Thái Nguyên.

23    In Bế Viết Đằng and Lục Văn Pào (1992), *passim*. Each of the stages in the Pụt Tày text contains a certain amount of what seems at first to be adventitious material. A full analysis would require comparison with similar texts from the same locality and other texts from elsewhere.

24    Original editor's note: A type of very hard wood usually used to make the handles of cleavers.

25    Editor's note: Probably this term refers to China. This word derives from *Khách* to define the Chinese. It is used as a general term referring to the land or country extending beyond the borders.

26    Editor's note: This tree has leaves that are used to dye the fabric used for religious rituals yellow.

27    On the matchlock guns used in pre-modern armies see Holm and Meng (2021, pp. 33–35).

28    This refers to the flowers of the banana.

29    On which see Holm (2004).

30    The Nùng are frequently referred to as working as hired hands in fields owned by the Tày. See Holm (2019).

31    The duty of reporting for such service is similar to the corvée duties of villagers and commoners under the rule of the chiefly domains or imperial rule. For a description of this system see Holm and Meng (2021, pp. 8–9, 18, 32–33); Took (2006, pp. 107–9, 201–6).

32    See Eliade (1964, p. 404); Rg-Veda III, 8, 5, tr. R. T. H. Griffith, II: 4. See also Holm (2018, p. 30).

33    See Eliade (1964, pp. 442–44; Eggeling 1882–1900, V: 2, 1, 10); see also Torcinovich (1999, pp. 237–49).

34    This ritual is found as an initiatory rite among the Manchu-speaking Xibo people in Xinjiang, among the Bouyei and other Tai-speakers in Guizhou, among the Jingpo, Yi and Lisu peoples in western Yunnan, among the Karen in Burma and the Black Thai in Laos, and in various forms of vernacular Taoist initiation. See Holm (2018, pp. 29–30). In scholarship on Chinese religion the Indic origins of this rite are seldom recognised.

35    Transcribed and translated in Holm (2018, Appendix pp. 56–61).

36    This was an epic compiled in the period between the fourth century BCE and the fourth century CE. See Sadakata Akira (1997, p. 26); Holm (2018, p. 39).

37    The exact dimensions of this world, including the breadth and depth of the seas, are set out in considerable detail in the *Abhidarmakośa*. See Pruden (1989, 2: 464 ff). Discussed also in Holm (2018, pp. 38–41).

38    This is of course a matter that requires further exposition. Meanwhile, for the circle of wind, see lines 215–222 in the Trùng Khánh version of the Khảm Hải, pp. 40–42 in Hoàng Triều Ân's edited version (*Tổng tập truyện thơ nôm các dân tộc thiểu số Việt Nam 2011*, vol. 11).

39    Michel Strickmann in his study on Tantric Buddhism in China (1996, p. 425) cites a range of studies documenting the underlayer of ancient elements in religious practices 'supposé indigènes et primitifs' among the Batak in Sumatra and the Cham and Bahnar in central and southern Vietnam.

40    See Holm (2018, p. 22) for the original text and a translation of the relevant passage.

41    Discussed in detail in Holm (2018, pp. 28–38).

42    Holm (2018, p. 25), citing Mus (1975, p. 24) to the effect that the Brāhmaṇas dated "from no earlier than the 7th century BC." This means that they were already well-established as part of the Vedic-Brahmanic tool-kit by the date of the earliest voyages from India to the east, and started to reach a critical mass by the early centuries of the Common Era. Mus (1975) is an English-language translation of Mus (1933).

43    Holm (2018, pp. 25–26). This phenomenon is what Paul Mus referred to as an 'extreme ritualistic ontology'.

44    Holm (2018, pp. 23–25). This is corroborated by a recent article by Taylor (2018), in which he adduces abundant evidence to show that much of the earliest infusion of Buddhism in the Jiaozhi area was in fact Brahmanical.

45    Wheatley (1983, pp. 286–310) cited in Holm (2018, p. 25).

46    Discussed in more detail in Holm (2018, pp. 17–21).

47    On the pre-Austronesian origins of seafaring in this region see Mahdi (2017).

48    Sather (2001). The group Sather investigated were the Saribas Iban in Borneo (p. 5), an Austronesian-speaking people. The interlinear text is found on pp. 203–683.

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
