# Peer review of "Roads to the Sky: Indic Ritual Elements in the Vietnam-China Borderlands and Their Maritime Transmission"

_religions, doi:10.3390/rel15121551_

Round 1

Reviewer 1 Report

Comments and Suggestions for Authors

This paper shows an excellent research on Pụt and Then rituals  among the Tày and Nùng peoples in northern Vietnam and southern China. The paper is perfectly structured and the analysis is appropriate. This field is rarely worked by scholars and this research is an important contribution to the academia.

According to their/his/her analysis of Tay shamanic songs and texts, the author(s) has(have) list(s) many stages in the journey to the sky and argue(s) that these stages “in the journey to the sky” demonstrate “that this Then ritual is an archetypical shamanic ritual.” Apparently, in this paper, the author(s) is(are) following Eliade’s classical trance theory which holds that shamanism is characterized by the soul travel to the other worlds rather than spirit possession. As the author(s) state(s) at the beginning of the paper, “these rituals are fundamentally different from those conducted in the same region by spirit mediums, which involve calling deities or departed ancestors down into the body of practitioner.” To my mind, this artificial categorizing was not followed by many other scholars. For example, I. M. Lewis (1971. Ecstatic Religion) argues that the shamanic trance includes both soul travel and spirit possession, and his argument is accepted by more scholars. I suggest that the author(s) should clarify why this Eliade’s statement is not outdated and still important today.

Although, the research and findings are original. Overall,rituals among the Tày and Nùng peoples are rarely studied by scholars and this research is an important contribution to the field. Therefore, I strongly suggest that this paper should be published. 

Line 139. “What I have done thus far is prepare…” I am wondering if “prepare” should be “preparing”.

I suggest that this paper should be published. 

Author Response

This paper shows an excellent research on Pụt and Then rituals  among the Tày and Nùng peoples in northern Vietnam and southern China. The paper is perfectly structured and the analysis is appropriate. This field is rarely worked by scholars and this research is an important contribution to the academia.

According to their/his/her analysis of Tay shamanic songs and texts, the author(s) has(have) list(s) many stages in the journey to the sky and argue(s) that these stages “in the journey to the sky” demonstrate “that this Then ritual is an archetypical shamanic ritual.” Apparently, in this paper, the author(s) is(are) following Eliade’s classical trance theory which holds that shamanism is characterized by the soul travel to the other worlds rather than spirit possession. As the author(s) state(s) at the beginning of the paper, “these rituals are fundamentally different from those conducted in the same region by spirit mediums, which involve calling deities or departed ancestors down into the body of practitioner.” To my mind, this artificial categorizing was not followed by many other scholars. For example, I. M. Lewis (1971. Ecstatic Religion) argues that the shamanic trance includes both soul travel and spirit possession, and his argument is accepted by more scholars. I suggest that the author(s) should clarify why this Eliade’s statement is not outdated and still important today.

Although, the research and findings are original. Overall,rituals among the Tày and Nùng peoples are rarely studied by scholars and this research is an important contribution to the field. Therefore, I strongly suggest that this paper should be published.

Line 139. “What I have done thus far is prepare…” I am wondering if “prepare” should be “preparing”.

I suggest that this paper should be published.

Reviewer 2 Report

Comments and Suggestions for Authors

The Author should update the bibliography concerning the ethnography and the history of northern Vietnam (Christian Culas, Jean Michaud, Christian Lenz), the classification policy of ethnic minorities and its evolution (Oskar Salemink, Masako Ito). Quoting Diguet doesn't seem really relevant anymore or if so please explain (please note that the book has been published in 1908 and not 1907: footnote vi).

This book could also be useful: Culas, Robinne (eds.) Inter-Ethnic Dynamics in Asia. Considering the Other through Ethnonyms, Territories and Rituals, Routledge, 2010.

The same for the Possesion cults and their relation with shamanism in North Vietnam ethnic Kinh: have a look on Paul Sorrentino: A l'epreuve de la possession, 2018.

page 3: the reader would like to know more about the manuscript. Can we speak about an "epopee"?

The author could explain and justify more clearly why he decided to quote precisely the extracts of the poem.

The author could change "Red River valley" with "Red river delta"

Author Response

Comments:

Comments and Suggestions for Authors

Reviewer: The Author should update the bibliography concerning the ethnography and the history of northern Vietnam (Christian Culas, Jean Michaud, Christian Lenz), the classification policy of ethnic minorities and its evolution (Oskar Salemink, Masako Ito). Quoting Diguet doesn't seem really relevant anymore or if so please explain (please note that the book has been published in 1908 and not 1907: footnote vi).

Response: Thank the reviewer for his/her suggestions and correction. I am actually reasonably up to date with Vietnamese scholarship on the Tay and Nung. The sources listed in the reviewer's suggestions above may be tangentially relevant, and I will check this more carefully in the future, but the topics of inter-ethnic relations, of highland peoples, and the operations of Vietnamese ethnic classification procedures are all not strictly relevant to the discussion in this article (the Tay are not a high mountain people like those covered in these sources). Contemporary developments likewise. Other works suggested - Salemink on the ethnic minorities of the Central Highlands are about a different area. The reason I chose to cite Diguet was precisely because his study was written in a period before all of the convulsive changes of the modern and contemporary period, and thus represented a status quo ante. For a more up-to-date review of the Nung, I could have consulted Cac dan toc o Viet Nam volume 2 (2016), but it was not necessary to do so.

Reviewer: This book could also be useful: Culas, Robinne (eds.) Inter-Ethnic Dynamics in Asia. Considering the Other through Ethnonyms, Territories and Rituals, Routledge, 2010.

Response: Thank you for this suggestion. The editors' names I take it are Christian Culas and François Robinne. Please see above comment.

Reviewer:

The same for the Possesion cults and their relation with shamanism in North Vietnam ethnic Kinh: have a look on Paul Sorrentino: A l'epreuve de la possession, 2018.

Response: The full title of this work is 'A l'epreuve de la possession - chronique d'une innovation rituelle dans le Vietnam contemporain'. Thank you for this suggestion, which I look forward to following up. This work would appear to deal primarily with contemporary developments, and therefore not be of direct relevance for the topic of this article. For spirit mediumship, of course, there is a massive amount of published scholarship. Many more titles could be added to the list. The main focus of the present article was on shamanic journeying.

Reviewer: page 3: the reader would like to know more about the manuscript. Can we speak about an "epopee"?

Response: Full information about the manuscript has been added. Thank you for this suggestion.

Reviewer: The author could explain and justify more clearly why he decided to quote precisely the extracts of the poem.

Response: The answer is that passages were chosen in order to fully justify the basic argument, and also demonstrate the degree of indigenisation.

Reviewer: The author could change "Red River valley" with "Red river delta"

Response: This may not always be appropriate, given that the area under discussion is wider than just the delta area.